# Widespread emergence of OmpK36 loop 3 insertions among multidrug-resistant clones of *Klebsiella pneumoniae*

**Sophia David**[1☯], **Joshua L. C. Wong**[2☯], **Julia Sanchez-Garrido**[2], **Hok-Sau Kwong**[3,4], **Wen Wen Low**[2], **Fabio Morecchiato**[5], **Tommaso Giani**[5,6], **Gian Maria Rossolini**[5,6], **Stephen J. Brett**[7], **Abigail Clements**[2], **Konstantinos Beis**[3,4], **David M. Aanensen**[1], **Gad Frankel**[2]*

**1** Centre for Genomic Pathogen Surveillance, Big Data Institute, Li Ka Shing Centre for Health Information and Discovery, University of Oxford, Oxford, United Kingdom, **2** MRC Centre for Molecular Bacteriology and Infection, Imperial College London, London, United Kingdom, **3** Rutherford Appleton Laboratory, Research Complex at Harwell, Didcot, Oxfordshire, United Kingdom, **4** Department of Life Sciences, Imperial College London; London, United Kingdom, **5** Department of Experimental and Clinical Medicine, University of Florence, Florence, Italy, **6** Clinical Microbiology and Virology Unit, Careggi University Hospital, Florence, Italy, **7** Department of Surgery and Cancer, Section of Anaesthetics, Pain Medicine and Intensive Care, Imperial College London, London, United Kingdom

☯ These authors contributed equally to this work.
* g.frankel@imperial.ac.uk

**Data Availability Statement:** All genome data used in this study is publicly available in Pathogenwatch (https://pathogen.watch/genomes/all?genusId=

## Abstract

Mutations in outer membrane porins act in synergy with carbapenemase enzymes to increase carbapenem resistance in the important nosocomial pathogen, *Klebsiella pneumoniae* (KP). A key example is a di-amino acid insertion, Glycine-Aspartate (GD), in the extracellular loop 3 (L3) region of OmpK36 which constricts the pore and restricts entry of carbapenems into the bacterial cell. Here we combined genomic and experimental approaches to characterise the diversity, spread and impact of different L3 insertion types in OmpK36. We identified L3 insertions in 3588 (24.1%) of 14,888 KP genomes with an intact *ompK36* gene from a global collection. GD insertions were most common, with a high concentration in the ST258/512 clone that has spread widely in Europe and the Americas. Aspartate (D) and Threonine-Aspartate (TD) insertions were prevalent in genomes from Asia, due in part to acquisitions by KP sequence types ST16 and ST231 and subsequent clonal expansions. By solving the crystal structures of novel OmpK36 variants, we found that the TD insertion causes a pore constriction of 41%, significantly greater than that achieved by GD (10%) or D (8%), resulting in the highest levels of resistance to selected antibiotics. We show that in the absence of antibiotics KP mutants harbouring these L3 insertions exhibit both an *in vitro* and *in vivo* competitive disadvantage relative to the isogenic parental strain expressing wild type OmpK36. We propose that this explains the reversion of GD and TD insertions observed at low frequency among KP genomes. Finally, we demonstrate that strains expressing L3 insertions remain susceptible to drugs targeting carbapenemase-producing KP, including novel beta lactam-beta lactamase inhibitor combinations. This study provides a contemporary global view of OmpK36-mediated resistance mechanisms in KP, integrating surveillance and experimental data to guide treatment and drug development strategies.

570&speciesId=573) and the European Nucleotide Archive (see S1 Table for accession numbers). Structural data corresponding to OmpK36$_{WT+TD}$ and OmpK36$_{WT+D}$ have been deposited in the Protein Data Bank with PDB ID codes 7PZF and 7Q3T, respectively.

**Funding:** SD and DMA are supported by funding from the National Institute for Health and Care Research (NIHR) Global Health Research Unit on Genomic Surveillance of Antimicrobial Resistance (NIHR 16/136/111; https://www.nihr.ac.uk). JLCW is supported by an MRC clinical PhD fellowship (MRC CMBI Studentship award MR/R502376/1; https://mrc.ukri.org). KB is supported by a grant from the MRC (MR/N020103/1; https://mrc.ukri.org). GF is supported by an Investigator grant from the Wellcome Trust (107057/Z/15/Z; https://wellcome.org). The funders played no role in the study design, data collection and analysis, decision to publish, or preparation of the manuscript.

**Competing interests:** The authors have declared that no competing interests exist.

## Author summary

Rapidly rising rates of antibiotic resistance among *Klebsiella pneumoniae* (KP) necessitate a comprehensive understanding of the diversity, spread and clinical impact of resistance mutations. In KP, mutations in outer membrane porins play an important role in mediating resistance to carbapenems, a key class of antibiotics. Here we show that resistance mutations in the extracellular loop 3 (L3) region of the OmpK36 porin are found at high prevalence among clinical genomes and we characterise their diversity and impact on resistance and virulence. They include amino acid insertions of Aspartate (D), Glycine-Aspartate (GD) and Threonine-Aspartate (TD), which act by decreasing the pore size and restricting entry of carbapenems into the bacterial cell. We show that these L3 insertions are associated with large clonal expansions of resistant lineages and impose a fitness cost evident during *in vivo* competition. Critically, strains harbouring L3 insertions remain susceptible to novel drugs, including beta lactam-beta lactamase inhibitor combinations. This study highlights the importance of monitoring the emergence and spread of strains with OmpK36 L3 insertions for the control of resistant KP infections and provides crucial data for drug development and treatment strategies.

## Introduction

*Klebsiella pneumoniae* (KP) is a leading cause of opportunistic infections in hospital and healthcare-associated settings worldwide [1,2]. Rates of antibiotic resistance among KP have risen rapidly in recent years, leading to its classification by the World Health Organisation as a critical priority resistant pathogen [3]. Of particular concern is the increasing number of KP infections that are resistant to carbapenems, which have been shown to be associated with a remarkable mortality burden [4]. Whilst newer agents with activity against carbapenem-resistant KP have been recently licensed [5], carbapenems remain vital in the treatment of severe infections due to their broad efficacy and limited adverse effects.

Carbapenem resistance in KP is primarily achieved by the acquisition of carbapenemase enzymes, which inactivate carbapenems by hydrolysis. These enzymes are typically plasmid-encoded and include variants of the KPC, OXA-48-like, NDM, VIM and IMP families [6]. Another important mechanism involves the modification of outer membrane porins which enable the non-selective diffusion of substrates, including both nutrients and antibiotics, into the bacterial cell [7,8]. These modifications restrict antibiotic entry and act in synergy with carbapenemase enzymes to increase the level of resistance. Resistance-associated mutations have been described in both major chromosomally-encoded KP porins, OmpK35 and OmpK36 [9,10]. In particular, truncations in the *ompK35* gene that result in a non-functional porin have been widely identified and are ubiquitous in a major high-risk clone comprising sequence types ST258 and ST512 [11,12]. By contrast, *ompK36* is rarely truncated and resistance mutations more commonly either reduce the abundance of OmpK36 in the outer membrane [13–15] or constrict the pore size [12,16]. Whilst OmpK35 truncations appear to have little to no effect on carbapenem resistance when present in isolation, they do appear to enhance the effect of resistance mutations in OmpK36 [16,17].

Mutations that mediate pore constriction have been shown to consist of amino acid insertions in extracellular loop 3 (L3) of OmpK36, a motif that conformationally determines the minimal pore radius. In particular, we previously determined the crystal structure of an OmpK36 variant (OmpK36$_{ST258}$, pdb 6RCP), identified from a KPC-2 expressing ST258 KP

isolate ($KP_{ST258}$), with a high minimum inhibitory concentration (MIC) to meropenem. $OmpK36_{ST258}$ contains a Glycine-Aspartate (GD) L3 insertion, which when introduced into a common OmpK36 variant, lacking an L3 insertion ($OmpK36_{WT}$, pdb: 6RD3), generating $OmpK36_{WT+GD}$ (pdb: 6RCK), led to a 10% reduction in minimal pore diameter. The substitution of the *ompK36* gene encoding $OmpK36_{WT}$ for $OmpK36_{WT+GD}$, in the genome of a laboratory KP strain, together with KPC-2 expression, phenocopied the carbapenem resistance profile of $KP_{ST258}$. Consistently, while the monosaccharide glucose (180g/mol) could freely diffuse across $OmpK36_{WT}$ and $OmpK36_{WT+GD}$, the diffusion of the larger disaccharide, lactose (342g/mol), was impaired in the presence of the L3 GD; this results in faster *in vitro* growth rates of strains expressing $OmpK36_{WT}$ when lactose was the sole carbon source present in minimal media [18].

We demonstrated that the extended L3 conformation in both $OmpK36_{ST258}$ and $OmpK36_{WT+GD}$ was stabilised by the formation of an intramolecular salt-bridge. *In silico* structural modelling has also predicted OmpK36 pore constriction by other L3 insertions (Threonine-Aspartate (TD) and Serine-Aspartate (SD)) observed in clinical KP genomes [12]. However, in the case of structural predictions, important derived metrics (e.g. minimal pore diameter) of these porin variants remain incomplete, precluding rational drug design. Moreover, studies assessing the prevalence of L3 insertions among clinical isolates have been restricted to the identification of *a priori* defined L3 insertions (i.e. GD/TD in Lam et al. 2021 [10]) and/or limited by the temporal and geographic breadth of available sample collections [10,12]. The increased availability of genomes now facilitates a more comprehensive analysis of different L3 insertions found among clinical KP worldwide, providing a valuable opportunity for informing surveillance, treatment and drug development strategies.

KP asymptomatically colonises mucosal surfaces including the gastrointestinal tract. It was detected in 3.8% of stool samples by 16S gene sequencing from healthy individuals in the Human Microbiome Project [19]. In addition, it is recognised that up to 80% of invasive infections, which underpin morbidity and mortality in patients, are caused by a patient's own gut colonising strain [20,21]. This is further exacerbated by high KP colonisation rates in hospitalised patients [21]. Whilst this enrichment may be multifactorial in nature, including precipitants such as systemic antibiotic administration leading to gut dysbiosis, it is unclear what precisely underpins the successful onward transmission of certain STs in healthcare environments.

Gut colonisation of KP, studied in murine models, revealed that KP expressing OmpK36 containing an L3 GD were outcompeted by isogenic strains expressing $OmpK36_{WT}$[17]. However, high KP inoculums ($10^{10}$ CFU per mouse) and continuous antibiotic treatment were needed to sustain the infection, indicating that mice do not appear to be good model hosts to investigate gastrointestinal colonisation dynamics.

Here we used a combination of bioinformatic and experimental approaches to detail the diversity, evolutionary dynamics and clinical impact of L3 insertions observed among KP isolates from a large global genome collection ($n = 16,086$). By solving additional OmpK36 structures, we show that other major L3 insertions beyond GD constrict the pore size and increase carbapenem resistance, and are associated with large clonal expansions among high-risk clones. These include Aspartate (D) and TD insertions in the important, albeit less well-studied, ST16 and ST231 lineages, respectively. We also demonstrate recurrent reversions of L3 insertions among clinical isolates, which we propose are underpinned by a competitive disadvantage in the absence of antibiotics. We provide evidence of this both *in vitro*, where lactose and amino acids/small peptide diffusion is abrogated, as well as *in vivo*, by competition in a preclinical mouse pneumonia model. Finally, we systematically evaluated the effect of D, GD and TD insertions on the susceptibility to novel antibiotic therapies, including key beta-

lactam/beta-lactamase inhibitor combinations, and show that these agents maintain efficacy despite pore constriction.

## Results

### D, GD and TD are the most common L3 insertions among clinical KP genomes

We investigated the prevalence of L3 insertions among a large collection of public KP genomes available in PathogenWatch (https://pathogen.watch/genomes/all?genusId=570&speciesId= 573; **S1 Table**). This collection comprises 16,086 assembled genomes from 84 countries, belonging to a total of 1,177 STs [22]. We unambiguously identified the *ompK36* gene in 94.4% (15,193/16,086) of the genomes; the gene was intact (i.e. not truncated) in 98.0% (14,888/15,193) of these. Among those with an intact *ompK36*, we found that 24.1% (3588/ 14,888) had one or more amino acids inserted into the L3 region (**Fig 1A**). A total of eight different L3 insertion types were observed, which comprised between one and three amino acids. 75.3% (2700/3588) of the L3 insertions observed were GD, while the remainder comprised TD (14.3%; 512/3588), D (7.8%; 281/3588), SD (2.0%; 73/3588), N (0.4%; 15/3588), TYD (0.1%; 4/ 3588), YGS (0.06%; 2/3588) and GG (0.03%; 1/3588) (**Fig 1B**).

L3 insertions were found in a total of 68 STs (GD—52 STs, TD—16 STs, D—15 STs, SD— 10 STs, N—1 ST, TYD—3 STs, YGS—2 STs), demonstrating their widespread emergence across the KP population. Notably, despite genetic redundancy, we found that the coding mutations for each L3 insertion were always the same (e.g. GD always encoded by ggc gac, TD by acc gac and D by gac). This homogeneity in the coding mutations may be at least partially explained by recombination, as evident from phylogenetic analysis of the *ompK36* open reading frame (ORF) that demonstrates some sharing of alleles between distantly related STs (https://microreact.org/project/52zjajyXDYr2ABfaaG2kaV-ompk36-gene-n14888). However, this analysis also showed parallel emergences of each insertion type across different *ompK36* gene backgrounds (**S1 Fig**). This may suggest an additionally important role for *de novo* mutation in the generation of L3 insertions; in addition, this implies that the particular underlying

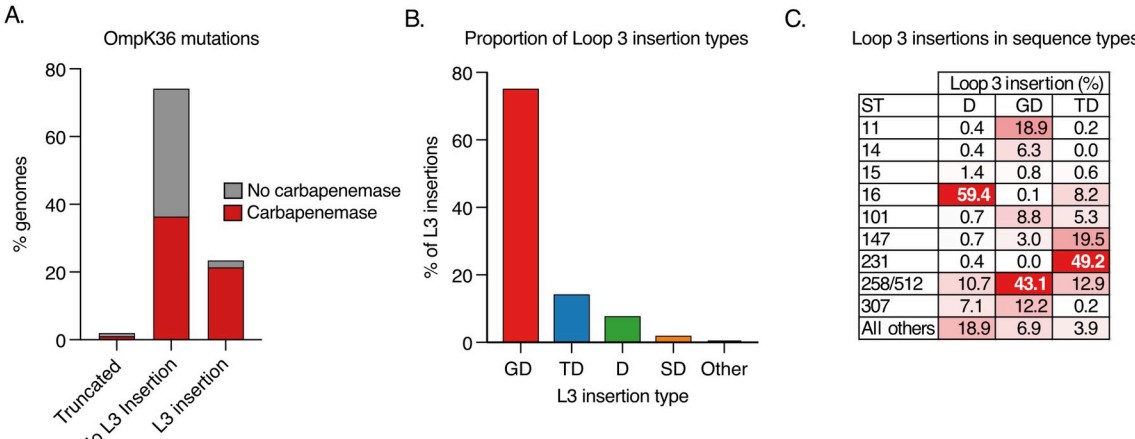

**Fig 1. Diversity of loop 3 (L3) insertions among a collection of 16,086 public KP genomes. A**. Proportion of genomes encoding an *ompK36* gene that is truncated or possessing a L3 insertion (if intact). Bars are coloured by the proportion of genomes carrying one or more carbapenemase genes. Only genomes with a single copy of *ompK36* that could be unambiguously characterised were included (*n* = 15,193). **B**. Proportion of L3 insertions identified that comprise amino acid insertions of Glycine-Aspartate (GD), Threonine-Aspartate (TD), Aspartate (D), Serine-Aspartate (SD) or others (N/GG/TYD/YGS). **C**. Distribution of genomes possessing each L3 insertion type across major high-risk sequence types (ST). ST258 and ST512 are grouped together as they form a single clone.

coding mutations of each L3 insertion type are more likely than the possible alternatives to evolve and/or spread.

In order to investigate the phenomenon of independent L3 insertion events being mediated by the same codon(s) and located at the same position within L3, we used the single D insertion, as it is encoded by a codon with a degeneracy of 2 (gac or gat). As in all clinical isolates D is encoded by gac we switched it to the synonymous gat codon on the KP genome (**S2A Fig**). In these strains we substituted the endogenous genomic *ompK36*$_{WT}$ ORF in KP strain ICC8001 with the mutant allele. This strategy was used to construct mutants throughout this work. We prepared outer membrane protein (OMP) fractions and following SDS-PAGE and Coomassie staining, found no visible difference in the OM abundance of OmpK36$_{WT+D}$ with gac or gat codons (**S2B Fig**). We then investigated if the position of the D insertion within L3 impacted on OM OmpK36 abundance. We generated a strain in which the D insertion was placed at -2 positions (-1 position does not exist as the D insertion immediately follows a D in the WT+D sequence and the amino acid sequence is therefore indistinguishable). OMP analysis demonstrated that the abundance of OmpK36$_{WT+D(-2)}$ was similar to OmpK36$_{WT+D}$ (**S2C Fig**). In contrast, insertion the D in the +1 position was not tolerated. This suggests that both D and GD insertions result from a duplication error of the preceding 3 or 6 nucleotides, respectively; however, this fails to explain why multiple insertions are all encoded by the same codons.

## D, GD and TD are found among global multi-drug resistant clones

We found that L3 insertions were mostly concentrated among global multi-drug resistant (MDR) clones (**Fig 1C**), implicating an important role in resistance. Altogether, 91.8% (3295/3588) were found in one of the top ten most frequently observed STs in the genome collection, which represent these major clones. This is despite genomes from these STs making up only 56.2% (9045/16,086) of the total collection. We also observed a high concentration of individual L3 insertions in particular clones. For example, 43.1% (1163/2700) of genomes with a GD insertion belonged to either ST258 or ST512 (which together make up the single clone, ST258/512), 49.2% (252/512) of genomes with TD belonged to ST231, and 59.4% (167/281) of genomes with D belonged to ST16 (**Fig 1C**). While all three clones are internationally dispersed, ST258/512 has largely been a dominant strain in the Americas, Europe and Middle East [23–27], and ST231 and ST16 are found at high prevalence in parts of Asia [28–30].

We also found that L3 insertions frequently co-occur with carbapenemase genes, which have also been shown to be concentrated among major high-risk clones [31]. Indeed, of the genomes in this collection with an L3 insertion, 90.7% (3255/3588) possessed one or more carbapenemases (**Fig 1A**). Furthermore, we observed that particular L3 insertions more frequently co-occur with some carbapenemases. For example, 76.0% (1854/2441) of carbapenemases found among genomes with a GD insertion were KPC, while 72.3% (334/462) and 75.7% (203/268) of those found among genomes with D and TD were from the NDM and OXA-48-like families, respectively. However, these associations are also confounded by the high concentration of particular carbapenemase genes among some lineages (e.g. KPC in ST258/512).

## L3 insertions are associated with clonal expansions in MDR lineages and revert at a low frequency

We next investigated the emergence and expansion patterns of L3 insertions among clinical genomes using a phylogenetic approach. We analysed the three MDR clones in which the D, GD and TD insertions predominate (D—ST16; GD—ST258/512; TD—ST231) (**Fig 1B**). We included genomes from the Pathogenwatch collection belonging to each of these STs, which

represented 3629 ST258/512 isolates (34 countries; collected 2003–2020), 446 ST16 isolates (26 countries; 2004–2020) and 302 ST231 isolates (19 countries; 2003–2019), and constructed a phylogeny of each after the removal of recombined regions using Gubbins.

The phylogenetic analysis of ST258/512 showed that an *ompK35* truncation and the KPC gene were largely ubiquitous and present in the earliest sampled isolates, while the lineage initially expanded in the absence of L3 insertions (**Fig 2A**). Since the emergence of this clone, L3 insertions have evolved many times independently. We found a total of six different L3 insertion types (D, N, GD, TD, SD, TYD) with D, GD and TD each emerging on multiple occasions. Several of the L3 insertion acquisition events (most notably of GD) are associated with subsequent clonal expansions. For example, the clade consisting largely of ST512 that encodes a GD insertion became highly successful as evident in the phylogeny and supported by multiple surveillance reports [23,32]. Our data also confirmed previous reports that this clade likely spread from the USA to Europe and the Middle East [31] where it dominated the resistant KP population in some countries over several years (e.g. Italy and Israel) [23,32]. Notably, the ST258/512 phylogeny also suggested that there have been multiple reversion events of L3 insertions, represented by genomes lacking a particular L3 insertion amidst a clade carrying that insertion (**Fig 2B**). The occurrence of reversions is suggestive of a selective pressure acting in favour of removing L3 insertions in certain contexts.

As with the ST258/512 lineage, our phylogenetic analysis of ST16 demonstrated that the lineage also expanded in the absence of L3 insertions and that these have since evolved frequently across different clades (**Fig 3A**). A total of five different L3 insertion types (D, GD, TD, SD, TYD) were found, with D, GD, TD and SD each evolving two or more times. We found a high diversity of carbapenemases among the ST16 lineage and numerous independent truncations of *ompK35*. However, most isolates (98.8%; 164/166) with the D insertion belonged to a single clade of isolates collected in Thailand between 2016–2018. This acquisition of the D insertion coincided closely with the gain of OXA-232 and NDM-1 carbapenemase genes and an *ompK35* truncation, followed by the rapid clonal expansion of this clade.

Contrary to the observations in ST258/512 and ST16, the ST231 phylogeny suggested that the TD insertion was acquired on a single occasion and associated with the major clade (**Fig 3B**). The low diversity within this clade is suggestive of a rapid clonal expansion. The acquisition of the TD insertion also coincided closely with the gain of OXA-232 and an *ompK35* truncation (with both likely occurring just prior to TD acquisition). No other L3 insertions were found in the ST231 lineage except for a single isolate with a D insertion. Phylogeographic analysis showed that the major clade encoding the TD insertion has spread to multiple countries, including India, Thailand and Oman, where significant local transmission is evident. As in the ST258/512 lineage, we also found numerous reversions of the TD insertion.

Finally, we investigated the role of recombination in the acquisition and reversion of L3 insertions in these three major lineages by assessing the recombined regions identified by Gubbins. While no recombination events involving *ompK36* were found in ST231, three were identified in ST258/512 and five in ST16. However, none of these events could be linked to either the acquisition or reversion of L3 insertions via a comparison of the affected clades. This would further imply a prominent role for *de novo* mutation, as previously suggested by the analysis of the global KP collection, although recombination involving closely-related strains or affecting only a small part of the gene (which Gubbins may be unable to detect) also cannot be ruled out.

## The L3 insertions reduce substrate diffusion and impact competitive fitness

We next aimed to define the extent to which the D and TD insertions, observed at highest prevalence after GD insertions, constrict the OmpK36 pore and increase meropenem

# A. ST258/512

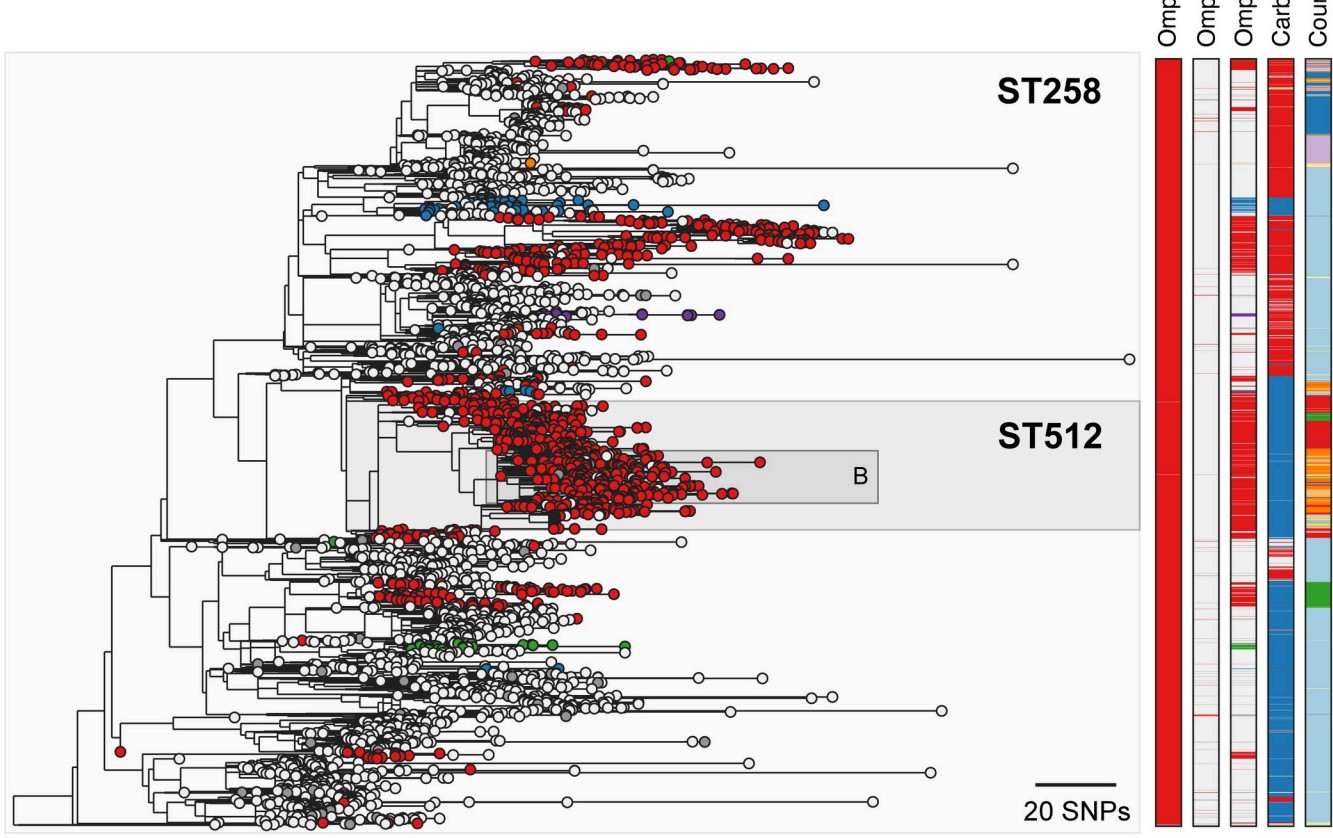

# B.

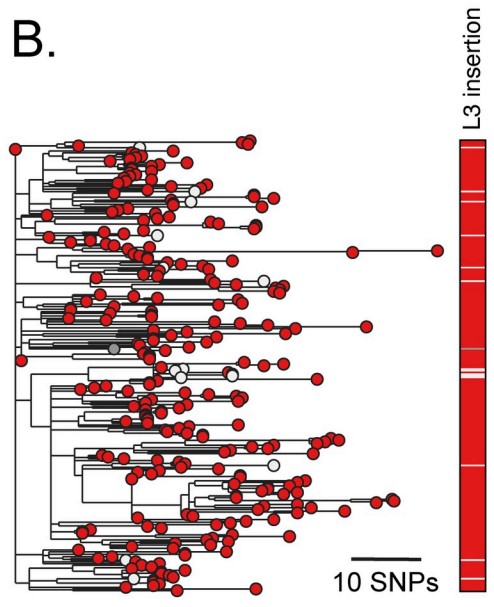

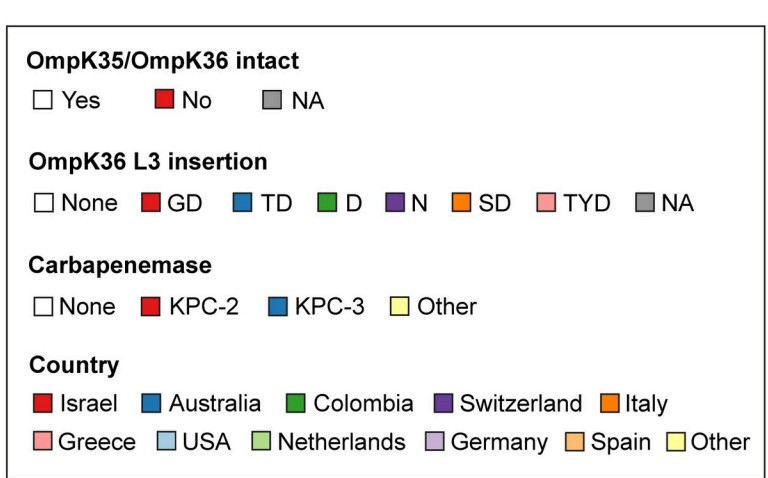

**Fig 2. Multiple acquisitions of GD and other L3 insertions among the high-risk ST258/512 clone. A.** Phylogenetic tree of 3629 isolates with public genome data belonging to sequence types (ST) 258 and 512. The tree was rooted using an ST895 isolate (accession SRR5385992) that was subsequently removed. Isolate tips are coloured by the type of OmpK36 L3 insertion (if applicable). Metadata columns from left to right show whether the *ompK35* and *ompK36* genes are intact (i.e., not truncated), the type of OmpK36 L3 insertion (if applicable), the carbapenemase gene type (if applicable) and the country of origin. Carbapenemases and countries are shown only for those with ≥15 isolates. Carbapenemase gene variants that have imperfect matches to known variants are grouped together with the most closely-related known variant. A similar interactive visualisation with more detailed metadata is available using Microreact at https://microreact.org/project/exB9brEAsQcpg7vKXMJtoF-k-pneumoniae-st258512 **B**. A zoomed-in visualisation of the clade highlighted in (A). Scale bars show the number of SNPs.

resistance. To that end we solved the crystal structures of chimeric OmpK36$_{WT}$ with D and TD insertions expressed in *E. coli* [18] (OmpK36$_{WT+D}$ and OmpK36$_{WT+TD}$) by X-ray crystallography and compared the minimal pore diameters to those of the previously solved OmpK36$_{WT}$ and OmpK36$_{WT+GD}$ structures (**Fig 4A–4D and S2 Table**). The OmpK36$_{WT+D}$ structure demonstrated the presence of two different L3 conformations, open or closed, both yielded similar minimal pore diameters (2.94 Å (D-open) and 2.95 Å (D-closed)). Notably, OmpK36$_{WT+TD}$ forms a particularly narrow channel with a minimal pore diameter of 1.88 Å. These values represent relative pore reductions of 8% (D) and 41% (TD) compared to OmpK36$_{WT}$, the latter significantly greater than that imposed by the GD insertion (10%).

To evaluate the effect of the reduced pore diameter on meropenem diffusion in OmpK36$_{WT+D}$ and OmpK36$_{WT+TD}$, we conducted liposome swelling assays. We also included OmpK36$_{WT}$ and OmpK36$_{WT+GD}$ variants, as they had been previously validated in these assays, and empty liposomes were used as a control to establish the baseline diffusion that occurs in the absence of OmpK36 channels. Diffusion rates were calculated by assessing changes in OD$_{400nm}$ per unit time ($\Delta OD_{400}/t(s)$). We found that liposomes with the OmpK36$_{WT+D}$ and OmpK36$_{WT+TD}$ variants had significantly reduced meropenem diffusion compared to OmpK36$_{WT}$, with rates similar to those observed with OmpK36$_{WT+GD}$ and empty liposomes (**Fig 4E**). We also measured the diffusion rate of glucose, a key carbon source, which has a substantially lower molecular weight than meropenem (180.2g/mol vs 383.5g/mol). We found similar glucose diffusion in the presence of all OmpK36 variants (**Fig 4F**), which was higher compared to empty liposomes.

In order to extend our understanding of the impact of the different OmpK36 L3 insertions on the diffusion of substrates we carried out a series of *in vitro* competition assays, in which isogenic KP strains expressing OmpK36 D (KP36$_{WT+D}$), GD (KP36$_{WT+GD}$) or TD (KP36$_{WT+TD}$) L3 insertions were competed against strains expressing OmpK36$_{WT}$ (KP36$_{WT}$). The competing KP strains were genomically tagged with either *sfGFP* or *mRFP1*. We inoculated a base solution of M9 minimal media, which was supplemented with a nutrient of interest. When glucose was added as the sole carbon source KP36$_{WT}$ did not outcompete any L3 insertion expressing strains (**Fig 4G–4I**). However, when this competition assay was repeated in the presence of the higher molecular mass disaccharide, lactose, KP36$_{WT}$ outcompeted all L3 insertion expressing strains. Moreover, repeating the experiment in the presence of cas-amino acids and glucose (as a carbon source) we found significant out competition of KP36$_{WT+D}$, KP36$_{WT+GD}$ and KP36$_{WT+TD}$ by KP36$_{WT}$ (**Fig 4G–4I**). As casamino acids is composed of amino acids and very small peptides [33], this data suggests these substrates also diffuse via OmpK36 and that this process, in turn, is impeded by the presence of L3 insertions.

We next used the isogenic strains expressing alleles encoding Ompk36$_{WT+D}$, OmpK36$_{WT+GD}$ and OmpK36$_{WT+TD}$ to determine the effect of the different L3 insertions on meropenem MIC. As a control, we generated a strain lacking *ompK36*, the deletion of which has been shown to increase carbapenem resistance [9]. Given the high frequency of *ompK35* truncations among high-risk clones we deleted the *ompK35* gene from all isogenic strains, and introduced the KPC-2-encoding plasmid, pKpQIL, by conjugation (see **Table 1** for list of strains and

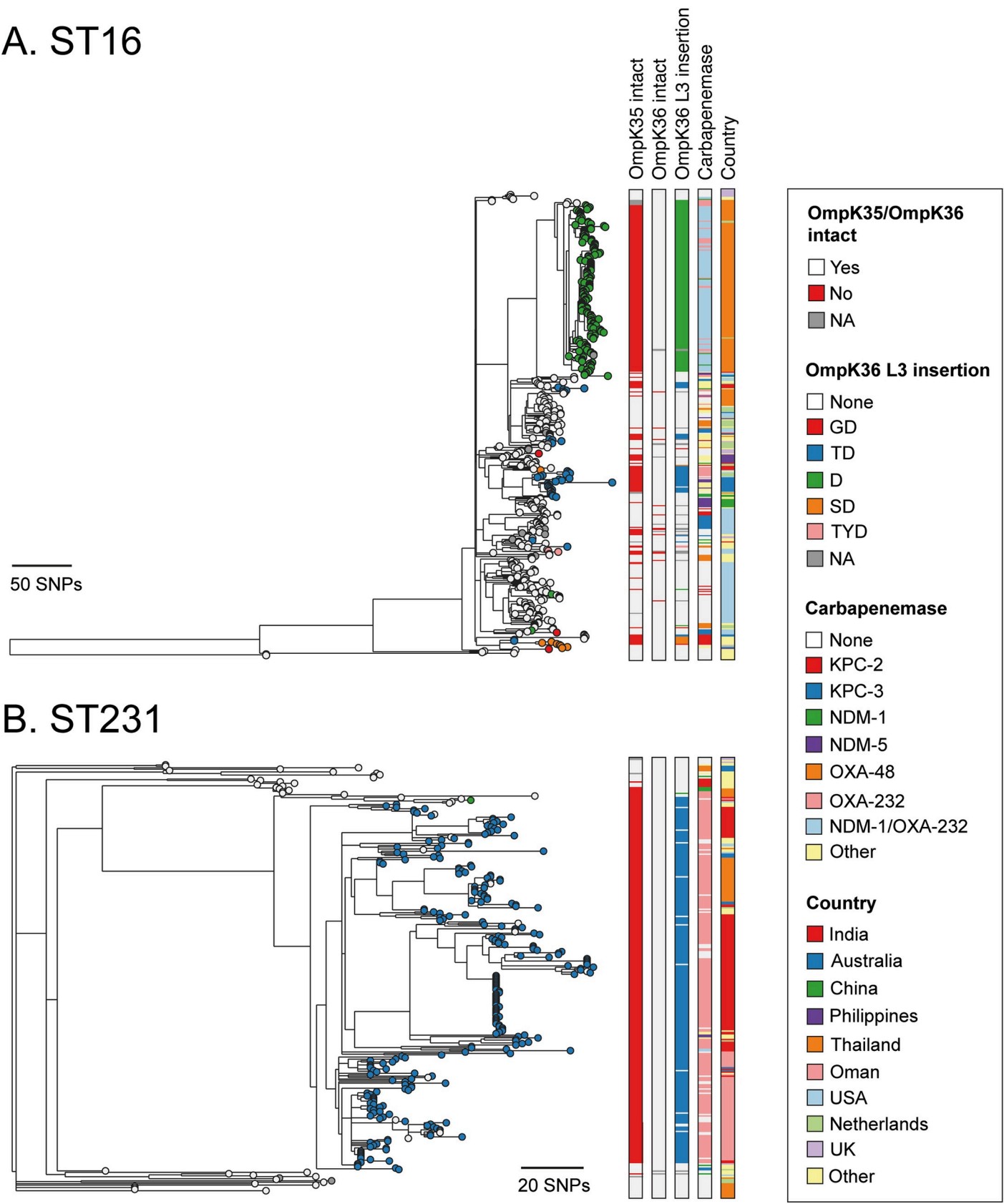

**Fig 3. The D and TD insertions are associated with large clonal expansions in ST16 and ST231.** Phylogenetic trees of 446 and 302 isolates with public genome data belonging to ST16 (**A**) and ST231 (**B**), respectively. The trees were rooted using an ST17 outgroup (accession ERR1228220) and an ST101 outgroup (accession ERR1216956), respectively, each of which were subsequently removed. Isolate tips are coloured by the type of OmpK36 L3 insertion (if applicable). Metadata columns from left to right show whether the *ompK35* and *ompK36* genes are intact (i.e., not truncated), the type of OmpK36 L3 insertion (if applicable), the carbapenemase gene type (if applicable) and the country of origin. Carbapenemases and countries are shown only for those with ≥10 isolates. Carbapenemase gene variants that have imperfect matches to known variants are grouped together with the most closely-related known variant. Scale bars show the number of SNPs. Similar interactive visualisations with more detailed metadata are available using Microreact at https://microreact.org/project/m8qd8j1YmfMapiPJ7prAEh-k-pneumoniae-st16 (ST16) and https://microreact.org/project/pZRm6DsxvZVYPQ2Ea33Buw-k-pneumoniae-st231 (ST231).

attributes). All L3 insertions increased the meropenem MIC four-fold from 16mg/L in *ompK36*$_{WT}$ to 64mg/L (the resistance breakpoint is >8mg/L) (**Fig 4J**). The absence of *ompK36* increased the MIC 32-fold to 512mg/L.

## KP expressing OmpK36 L3 insertions display a competitive disadvantage *in vivo*

The high prevalence of the different OmpK36 L3 insertions across the KP population, together with the observation that reversions also occur, led us to explore the impact of the D, GD and TD L3 insertions on bacterial fitness during invasive severe infection using a mouse pneumonia model. To do this, we performed *in vivo* infection experiments using strains encoding either *ompK36*$_{WT}$ (KP36$_{WT}$), *ompK36*$_{WT+D}$ (KP36$_{WT+D}$), *ompK36*$_{WT+GD}$ (KP36$_{WT+GD}$) or *ompK36*$_{WT+TD}$ (KP36$_{WT+TD}$) (**Table 1**). Intratracheal intubation was used to inoculate 250 CFU of KP directly into the lungs of mice, replicating the mode of infection in ventilator-associated pneumonia (**Fig 5A**). A control group of animals received PBS alone. After 48 h, infection with all isogenic strains induced significant weight loss compared to those receiving PBS only (**Fig 5B**). However, no significant differences were observed between those infected with KP36$_{WT}$ or KP36$_{WT+D}$, KP36$_{WT+GD}$ or KP36$_{WT+TD}$. Similarly, all isogenic strains achieved high CFU counts in the lungs and blood (although not all animals were bacteraemic at the end of the time-course), with no significant differences observed between groups (**Fig 5C and 5D**). Measurement of proinflammatory cytokines revealed significant increases of serum IL-6 and CXCL-1 following infection with any strain compared to the PBS control, but with no significant differences observed between the strains themselves; however, serum TNF was only significantly elevated in KP36$_{WT}$ infection compared to the uninfected controls (**Fig 5E–5G**). Lastly, we found significant increases in lung neutrophils induced by all infecting strains compared to the PBS control, but again observed no significant differences between the four OmpK36 backgrounds (**Fig 5H**). These experiments suggest that the L3 insertions do not significantly attenuate KP infection, thereby explaining the successful clonal expansions observed among isolates carrying these mutations.

We next used a more stringent method of assessing relative bacterial fitness by competing KP strains with L3 insertions against KP36$_{WT}$ *in vivo* in the lungs. We infected mice with a total inoculum of 500 CFU, comprising of 50% KP36$_{WT}$ and 50% of either KP36$_{WT+D}$ or KP36$_{WT+TD}$ (KP36$_{WT+GD}$ having been tested previously [16]) (**Fig 5I**). To identify the strains at the experimental end-point (36 hpi), we used chromosomal *sfGFP* and *mRFP1* tags. We enumerated lung CFU counts as the outcome measure and found that both KP36$_{WT+D}$ and KP36$_{WT+TD}$ (as with KP36$_{WT+GD}$ previously) were outcompeted by KP36$_{WT}$ (**Fig 5J–5L**). These findings demonstrate a competitive disadvantage, at least during invasive infection in the lungs, in the absence of antibiotics. These findings suggest that competitive disadvantage in specific contexts (i.e. absence of antibiotics) may explain the reversions of L3 insertions in the KP population.

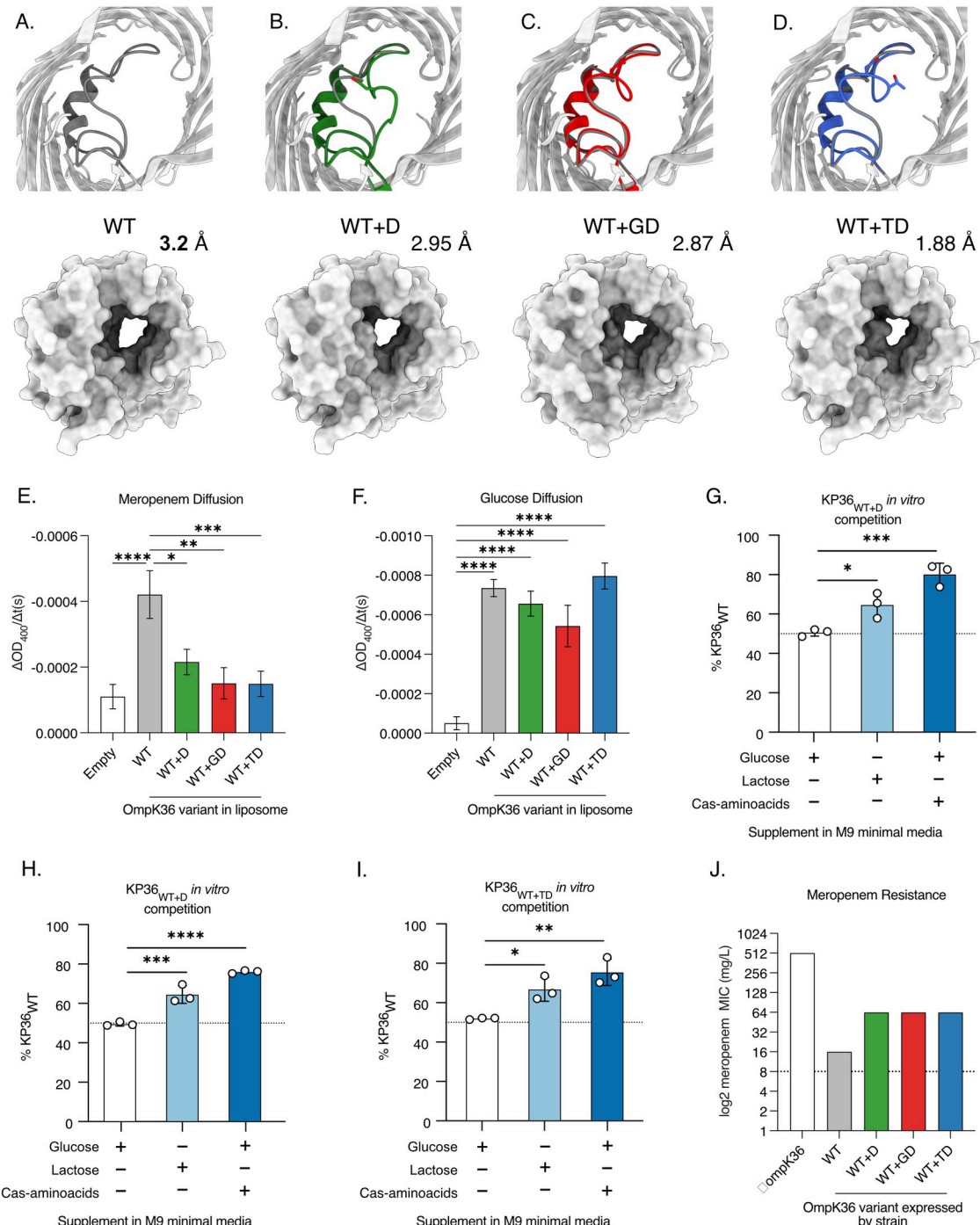

**Fig 4. L3 insertions reduce OmpK36 pore diameter and restrict the diffusion of meropenem. A-D.** Cartoon illustrations of OmpK36 (grey) in which L3 is coloured for all the variants; the view is from the extracellular space and perpendicular to the membrane (top panels). Insertions in L3 reduce the relative pore radius as calculated by the program HOLE [58]; surface representation to show the impact of the mutations in the pore size (bottom panels). **E-F.** Rate of diffusion ($\Delta OD_{400}/t(s)$ of meropenem (E) and glucose (F) as determined by liposome swelling assays for different OmpK36 variants. **G-I:** *In vitro* competition against KP36$_{WT}$ and KP36$_{WT+D}$ (**G**), KP36$_{WT+GD}$ (**H**) and KP36$_{WT+TD}$ (**I**). No competitive advantage is observed when glucose is added to the M9 minimal media. Lactose and cas-aminoacids supplementation results in outcompetition of L3 insertion expressing strains by KP36$_{WT}$. **J.** Median meropenem MIC values for strains encoding different *ompK36* variants ($n$ = 3 replicates). **E-F:** All comparisons not shown were non-significant. Error bars ±SEM. $^{*}$ p<0.332; $^{**}$ p<0.0021, $^{***}$ p<0.0002, $^{****}$ p<0.00001, statistical significance determined by ordinary one-way ANOVA with Tukey's multiple comparison test. **G-I: Error bars** ±SD. $^{*}$, $^{*}$ p<0.332; $^{**}$ p<0.0021, $^{***}$ p<0.0002, $^{****}$ p<0.00001, statistical significance determined by ordinary one-way ANOVA with Dunnett's multiple comparison to glucose supplementation with a single pooled variance.

**Table 1. Isogenic strains used in this study.** The parental strain ICC8001 [16] was genetically modified in a seamless and markerless recombineering approach to generate the *ompK36* variants.

| Strain | *ompK35* | *ompK36* |
|---|---|---|
| KPΔ36 | Δ | Δ |
| KP36_WT | Δ | Wild-type (no loop 3 insertion) |
| KP36_WT+D | Δ | Loop 3 Aspartate (D) insertion (codon = GAC) |
| KP36_WT+GD | Δ | Loop 3 di-amino acid Glycine-Aspartate (GD) insertion |
| KP36_WT+TD | Δ | Loop 3 di-amino acid Threonine-Aspartate (TD) insertion |
| KP36_WT+D(-2) | Δ | Loop 3 Aspartate (D) insertion in position -2 relative to the naturally occurring D insertion |
| KP36_WT+D(GAT) | Δ | Loop 3 Aspartate (D) insertion (codon = GAT) |

## Novel drugs targeting KPC-producing KP are effective against L3 insertion-expressing strains

Finally, we used our isogenic strain collection to systematically test the impact of D (KP36_WT+D), GD (KP36_WT+GD) and TD (KP36_WT+TD) insertions on susceptibility to new or recently licensed antibiotic therapies that are vital for the treatment of carbapenemase-producing KP (**Table 2**). We evaluated four beta-lactam/beta-lactamase inhibitor combinations: ceftazidime/avibactam (CAZ/AVI), meropenem/vaborbactam (MER/VAB), imipenem/relebactam (IMI/REL) and aztreonam/avibactam (AZT/AVI)) and the novel siderophore cephalosporin cefedericol (FDC). When combination drugs were assessed, we also determined the MIC in the absence of the beta-lactamase inhibitor (i.e., parental drug alone). All strains had *ompK35* deleted and expressed the KPC-2 carbapenemase. We also evaluated the impact of deleting *ompK36* (KPΔ36) to replicate the effect of *ompK36* truncation, which was observed, albeit rarely, in our genomic analyses (**Figs 2A and 3A and 3B**).

The MICs to the parental drugs used in co-formulations (MER, CAZ, IMP and AZT) were in the resistant range, irrespective of the OmpK36 variant expressed (**Table 2**). As already described for MER (**Fig 4G**), D, GD and TD insertions increased the MIC as compared to wild-type OmpK36 expression. Whilst the MICs to AZT were universally above the range of the assay, CAZ and IMP resistance was found to be inversely correlated with the minimal pore diameter imposed by the L3 insertion type, with the KP36_WT+TD strain achieving higher MICs than seen in KP36_WT+D and KP36_WT+GD. The KPΔ36 strain achieved the highest MICs for each drug with the exception of CAZ, where KP36_WT+TD obtained the highest level of resistance. When we tested these drugs in combination with their respective beta-lactamase inhibitors, susceptibility was restored among all strains harbouring L3 insertions. However, some increase was observed for the MICs of CAZ/AVI, MER/VAB and AZT/AVI (relative to KP36_WT), suggesting that L3 insertions have some effect on the susceptibility to these novel beta-lactamase-inhibitor combinations. Of note, KPΔ36 presented a resistant phenotype to IMP/REL, which was not observed with MER/VAB, CAZ/AVI or AZT/AVI. All strains were susceptible to FDC, in keeping with a porin independent uptake mechanism via siderophore receptors [34]. However, we noted that the FDC MIC was highest in KPΔ36 indicating that entry can, in part, be mediated by OmpK36. Taken together, these results show that novel drugs targeting carbapenemase-producing KP remain effective against strains possessing L3 insertions.

## Discussion

Monitoring the emergence, spread and clinical impact of resistance mutations among KP isolates is essential to informing public health intervention strategies. Here we demonstrate the

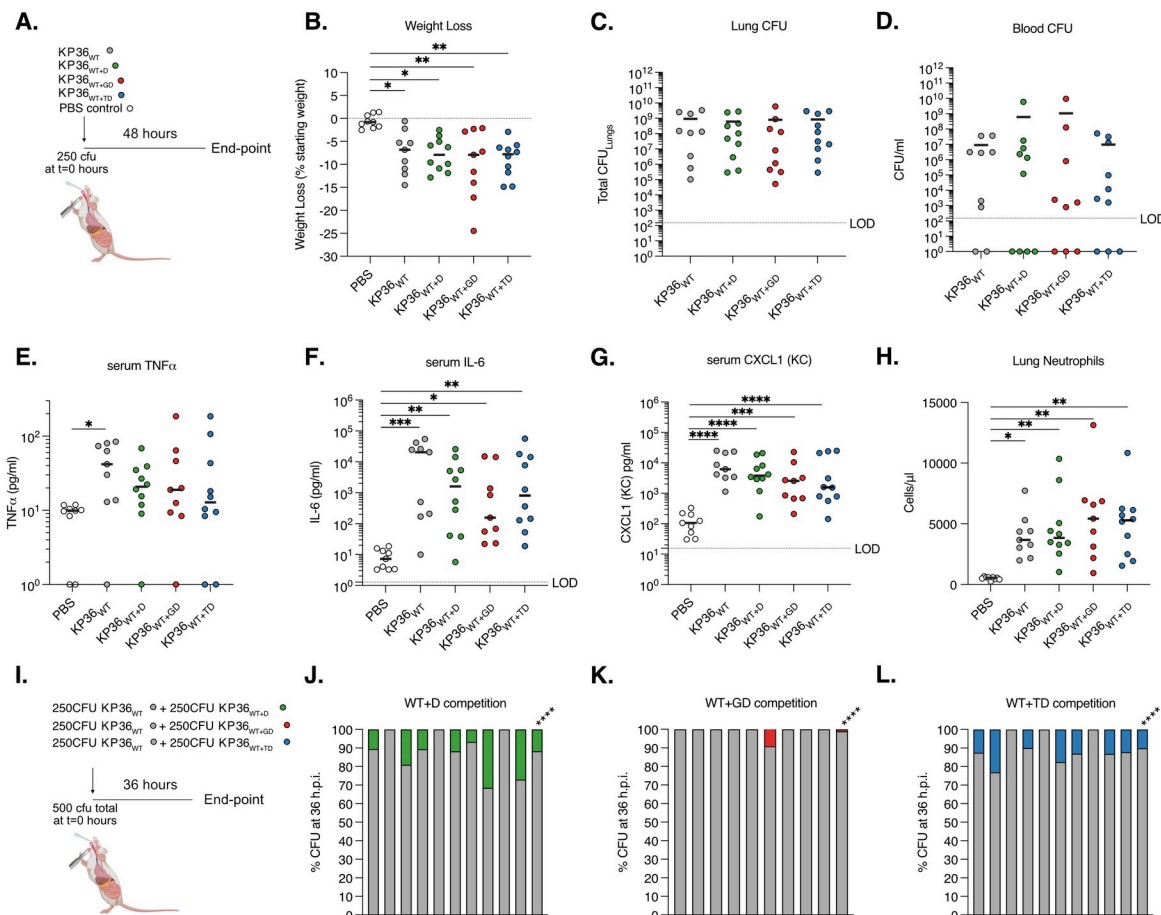

**Fig 5. OmpK36 L3 insertions maintain virulence but confer a competitive disadvantage in a preclinical murine pneumonia model.**
**A-H.** Pneumonia was induced by the intratracheal administration of 250 CFU of isogenic KP strains expressing D (KP36$_{WT+D}$), GD (KP36$_{WT+GD}$) and TD (KP36$_{WT+TD}$) OmpK36 L3 insertions. A strain lacking any L3 insertion (KP36$_{WT}$) and PBS (uninfected) were used as controls. A schematic of the infection is outlined in panel **A**. At 48 hours post infection significant weight loss was induced by infection with all strains, irrespective of OmpK36 variant (**B**) and no significant differences were observed between strains. No significant differences were observed in the lung (**C**) and blood (**D**) bacterial burdens between infection with any strain. Serum TNF was only significantly increased following infection with KP36$_{WT}$ compared to uninfected (PBS) controls (**E**). Serum IL-6 (**F**), CXCL-1 (**G**) and lung neutrophils (**H**) significantly increased following infection with all strains compared to uninfected (PBS) controls, with no significant differences between strains observed. **I-L.** Competition assays were employed to stringently assess the fitness of OmpK36 L3 insertion mutations. A schematic of the infection is outlined in panel **I**. 250 CFU of KP36$_{WT}$ was competed against 250 CFU of KP36$_{WT+D}$ (**J**), KP36$_{WT+GD}$ (**K**) or KP36$_{WT+TD}$ (**L**) and bacterial burdens were assessed at 36 hours post infection in the lungs. Each graph shows the % CFU recovered in the lungs in individual mice followed by a summary bar with the mean competition result across infections. KP36$_{WT}$ significantly outcompeted all the L3 insertions tested. *, p < 0.05; **, p < 0.01; ***, p < 0.001; ****, p<0.0001. All experiments were conducted in biological duplicate with 4–5 mice per group. **B-H** Significance was determined by ordinary one-way ANOVA followed by Tukey's multiple comparison post-test, except in **E** where Kruskal-Wallis test was employed as data was not normally distributed. **J-L** Mean competition was assessed by Mann-Whitney T-test. The diagrams in A and I were created with BioRender.com.

widespread distribution of OmpK36 L3 insertions among clinical KP isolates worldwide, as inferred from a large collection of publicly available genomes (*n* = 16,086). In particular, three types of L3 insertion, comprising amino acid insertions of D, GD and TD, made up 97.4% of those identified and were found in 23.5% (3493/14,888) of all genomes encoding an intact *ompK36* gene. Among genomes encoding one or more carbapenemase genes, this proportion increased to 36.1% (3171/8795). While there is a bias towards sequencing resistant isolates, our data nevertheless demonstrates that these mutations are one of the major adaptations of KP to antibiotic-rich healthcare environments.

**Table 2. Minimum inhibitory concentrations of different antibiotics for isogenic KP strains.**

| Strain | MIC (mg/L) | | | | | | | | |
|--------|-----|---------|-----|---------|-----|---------|------|---------|------|
|        | MER | MER/VAB | CAZ | CAZ/AVI | IMP | IMP/REL | AZT  | AZT/AVI | FDC  |
| KPΔ36  | **512** | 1 | **64** | 1 | **512** | **4** | >**1024** | 1 | 0.25 |
| KP36<sub>WT</sub> | **16** | 0.06 | **16** | 0.5 | **8** | 0.5 | >**1024** | 0.25 | 0.06 |
| KP36<sub>WT+D</sub> | **64** | 0.5 | **32** | 0.5 | **32** | 0.5 | >**1024** | 0.5 | ≤0.06 |
| KP36<sub>WT+GD</sub> | **64** | 0.25 | **64** | 1 | **32** | 0.5 | >**1024** | 0.5 | ≤0.06 |
| KP36<sub>WT+TD</sub> | **64** | 1 | **128** | 4 | **64** | 0.25 | >**1024** | 1 | ≤0.06 |

All strains express KPC-2 from a pKpQIL-like plasmid and contain a genomic *ompK35* deletion.

MER = meropenem, MER/VAB = meropenem/vaborbactam, CAZ = ceftazidime, CAZ/AVI = ceftazidime/avibactam, IMP = imipenem, IMP/REL = imipenem/relebactam, AZT = aztreonam, AZT/AVI = aztreonam/avibactam, FDC = cefiderocol. Values in bold and underlined represent MICs in the resistant range.

EUCAST breakpoints used are as follows: MER: S = ≤2; R = >8, MER/VAB: S = ≤8; R = >8, CAZ: S = ≤1; R = >4, CAZ/AVI: S = ≤8; R = >8, AZT: S = < = 1; R = >4, AZT/AVI: S = ≤1; R = >4 (AZT breakpoints used in the absence of a consensus for the combination), IMP: S = ≤2; R = >4, IMP/REL: S = ≤2; R = >2, FDC: S≤2; R = >2.

We solved the structures of OmpK36 with the D and TD insertions and compared their minimal pore diameters to those of the wild-type OmpK36 porin and OmpK36 with a GD insertion determined previously [16]. This revealed variation in the degree of pore constriction imposed by different L3 insertions, with reductions in pore size from the wild-type OmpK36 ranging from 41% (TD) to 10% (GD) to 8% (D). Interestingly, all three L3 insertions increased the meropenem MIC by the same magnitude (four-fold) compared to wild-type OmpK36 expression. However, the severity of pore constriction was reflected in the resulting MIC values for imipenem and ceftazidime (i.e. TD-expressing strain exhibiting the highest resistance). These differences are therefore vital in the process of rational physico-chemical drug design. We also found that L3 insertions have no effect on the diffusion of glucose, a key carbon source that is of lower molecular weight than beta-lactam antibiotics. The ability to maintain this physiological role of OmpK36 thereby demonstrates a key advantage of using pore constriction to impede antibiotic entry rather than mutations that reduce *ompK36* expression or result in a non-functional, truncated porin.

We showed that L3 insertions have emerged widely across the KP population and are most concentrated among known high-risk clones. In particular, ST258/512 accounted for a large proportion (43.1%) of GD insertions, while ST231 and ST16 harboured a high proportion of the TD (49.2%) and D (59.4%) insertions respectively. The relative lack of surveillance and availability of ST231 and ST16 genomes as compared to ST258/512 sequences can partially account for the under-recognition of the TD and D insertions to date, despite their high global clinical impact. Among these different clonal lineages, we found that L3 insertions often coincide with carbapenemases and *ompK35* truncations, and we found multiple instances of where the acquisition of these three traits in close succession was followed by rapid clonal expansion. Examples of this are the major clade of the ST231 lineage, associated with high transmission in India (and elsewhere) [28,30], as well as a large clonal expansion of an ST16 subtype in Thailand. We propose that these expansions are context dependent, with positive selection likely caused by the widespread use of carbapenems prior to the availability of novel drugs specifically targeting carbapenemase-producing *Enterobacteriaceae* (CPE). In particular, high-dosage carbapenems in combination with other drugs (e. g. colistin, tigecycline, fosfomycin) or even the use of double carbapenem regimens were mostly recommended for treatment of CPE infections [35,36]. Surveillance and infection control efforts must now focus on limiting spread of these resulting clones, as well as rapidly detecting the convergence of these resistance traits among other STs.

The ability of strains encoding L3 insertions to undergo large clonal expansion events fits with our finding that these mutations do not reduce the infection capacity of KP in a mouse pneumonia model. This contrasts with mutations resulting in non-functional OmpK36, which cause even higher carbapenem resistance but have been shown to result in significant attenuation *in vivo* [9,16] and rarely proliferate beyond individual patients. However, we did also find reversion events of L3 insertions (namely GD and TD) in clinical isolates, suggestive of a selection pressure acting to revert the pore to wild-type in certain contexts (e.g., in the absence of antibiotics). The *in vivo* competition experiments performed here and previously [16,17] demonstrate a competitive disadvantage of the three common OmpK36 L3 insertions relative to a wild-type OmpK36-expressing strains without antibiotic pressure. This suggests that antibiotic stewardship measures could play a crucial role in limiting further expansion of resistant KP carrying L3 insertions. Whilst a murine gastrointestinal competition model may add support to this hypothesis, it is inherently confounded by the requirement for antibiotic administration to maintain colonisation, precluding testing the precise context in which reversions may occur (i.e. the absence of said antibiotics). However, we do build on the role of OmpK36 in KP physiology, demonstrating diffusion of nitrogen sources, as well as carbon sources, through this pore; L3 insertions negatively impact of this diffusion and this potentially explains the marked outcompetition we observed in the host environment.

Finally, the high prevalence of L3 insertions among carbapenemase-producing strains led us to determine the precise impact of these mutations on the efficacy of new or recently licensed drugs targeted at this group of resistant KP. While relatively rare overall, cases of resistance emerging to the novel combination therapies (MER/VAB, CAZ/AVI, IMP/REL, AZT/AVI) during therapy have been reported [37]. Resistance typically involves mutations or increased expression of a beta-lactamase (including KPC and AmpC enzymes) [38]. However, loss or downregulation of OmpK36 has also been associated with increased resistance to MER/VAB [14], CAZ/AVI [39–41] and IMP/REL [40,42]. Here we found that the addition of the inhibitors to the parental drugs restores susceptibility in KPC-2 producing strains with OmpK36 L3 insertions, with MICs all below resistance breakpoints. Strains not expressing OmpK36 were also susceptible to all combination drugs, with the exception of IMP/REL. Overall, these findings suggest that beta-lactamase inhibitor entry is largely OmpK36-independent. However, the MICs of CAZ/AVI, MER/VAB and AZT/AVI were modestly affected by L3 insertions, suggesting a potential contribution to increasing resistance in the presence of other mechanisms. Similarly, we found that the siderophore cephalosporin (FDC) remains effective against strains with L3 insertions, in line with its primary entry via iron uptake receptors [34].

Taken together, our data highlights OmpK36 L3 insertions as a crucial priority in the global surveillance of carbapenem resistant KP due to their high prevalence among clinical isolates and associations with large clonal expansions. We also propose that as genomic surveillance becomes increasingly adopted, especially in low- and middle-income countries, monitoring the diversity of such resistance mechanisms over globally representative regions will be vital for optimisation of treatment and drug development strategies worldwide.

## Materials and methods

### Ethics statement

All animal work took place under the auspices of the United Kingdom Animals (Scientific Procedures) Act 1986 (License: PP7392693) and was locally approved by the institutional ethics committee (the Animal Welfare and Ethical Review Body [AWERB]).

## Identification and characterisation of *ompK36* among public genomes

We used a public collection of 16,086 KP genomes available in Pathogenwatch [22] (https://pathogen.watch/genomes/all?genusId=570&speciesId=573; accessed September 2021) to characterise the diversity of *ompK36* genes. The *ompK36* gene was identified in the short-read assemblies using BLASTn v2.6.0 [43] with a query gene from the reference genome, ATCC43816 (accession CP009208). To unambiguously identify *ompK36*, we required a single hit per assembly that matched ≥10% of the query length, possessed ≥90% nucleotide similarity and contained a start codon. Seaview v4.7 [44] was used to translate the nucleotide sequences to protein sequences using the standard genetic code. Non-truncated protein sequences (i.e. those with ≥95% of the query length) and the corresponding gene sequences were aligned using MUSCLE v3.8 [45]. These alignments were used to identify all intact protein and gene variants present. The variants were analysed together with the metadata and genotyping data (e.g., multi-locus sequence typing and resistome data) available in Pathogenwatch. A phylogenetic tree of all intact *ompK36* gene sequences was constructed based on the variable sites using RAxML v8.2.8 [46] and visualised using Microreact v166 [47].

## Phylogenetic analysis of ST258/512, ST231 and ST16 lineages

Raw sequence reads were downloaded from the European Nucleotide Archive (ENA) for all KP genomes belonging to STs 258/512 (*n* = 3673), 231 (*n* = 307) and ST16 (*n* = 453) in Pathogenwatch. Reads were mapped using Burrows Wheeler Aligner v0.7.17 [48] to a lineage-specific reference genome: NJST258_1 (accession CP006923) [49] for ST258/512, FDAARGOS_629 (accession NZ_CP044047) for ST231, and QS17_0029 (accession NZ_CP024038) for ST16. SNPs were identified using a pipeline comprising SAMtools mpileup v0.1.19 [50] and BCFtools v0.1.19, and pseudo-genome alignments were generated for each lineage. Individual genomes were excluded from subsequent analyses if the mean mapping coverage was <20x or if ≥25% of positions in the pseudo-genome alignment were missing data. Recombined regions were removed from the alignments and a phylogenetic tree was generated with the remaining variable positions using Gubbins v2.4.1 [51]. An outgroup isolate was also included in these analyses for each lineage in order to root the phylogenetic trees (SRR5385992 from ST895 for ST258/512, ERR1216956 from ST101 for ST231, ERR1228220 from ST17 for ST16). Phylogenetic trees were visualised together with all metadata and genotyping data using Microreact v166 [47].

## Generation of OmpK36 L3 insertion mutants

Genome editing took place in the laboratory KP strain ICC8001 (a derivative of ATCC43816) using the pSEVA612S system and lambda-red fragment mediated homologous recombination as previously described. The *ompK35* gene (open reading frame) deletion was carried out in previous work [15]. Mutagenesis vectors to generate the D and TD insertion, as well as the D insertion variants, were made by site directed mutagenesis using primers 1–7 (**S3 Table**). and PCR products were recircularised using KLD enzyme blend (New England Biolabs (UK)). Genome modifications were checked by sequencing of genomic *ompK36* locus PCR products generated using primers 8/9 and Sanger sequencing (Eurofins Genomics GmBH). sfGFP and mRFP1 were introduced into the silent *glmS* genomic site of strains generated in this manuscript as previously described [16].

## Outer membrane preparations and gel electrophoresis

Outer membrane proteins were purified using a previously described protocol with some modifications [52]. Briefly overnight cultures of bacteria were grown in LB (10g/L NaCl),

washed in 10 mM HEPES buffer (pH 7.4) and sonification was performed at 25% amplitude for 10 bursts of 15 seconds each. After removal of cellular debris via centrifugation, outer membrane proteins were pelleted by high speed centrifugation, washed in HEPES buffer supplemented with 2% sarcosine buffer and finally resuspended in water. 10 μg of protein were separated by SDS-PAGE using 12% acrylamide Mini-protean TGX precast gels (Bio-Rad, USA); gels were stained with Coomassie (Sigma-Aldrich).

## Purification of OmpK36 variants

OmpK36 variants were purified using our previously established protocol without any modifications [16]. All OmpK36 variants were purified in 10 mM HEPES pH 7, 150 mM NaCl and 0.4% $C_8E_4$.

## Crystallisation

OmpK36$_{WT+TD}$ and OmpK36$_{WT+D}$ were exchanged into 10mM HEPES pH 7, 150mM LiCl, and 0.4% $C_8E_4$ prior to crystallisation. Crystals for both variants were grown from a solution containing 0.1M Lithium sulfate, 0.1M sodium citrate pH 5.6 and 12% PEG4000 at 20˚C. The crystals were cryoprotected by supplementing the crystallisation condition with 25% ethylene glycol and were frozen in liquid nitrogen. Diffraction screening and data collection were performed at Diamond Light Source synchrotron.

## Data collection and structure refinement

OmpK36$_{WT+D}$ data to 1.78 Å were collected on I24 at Diamond Light Source at a wavelength of 0.97 Å using a Pilatus3 6M detector and processed using xia2 [53]. The space group was determined to be *P1* with six copies of OmpK36$_{WT+D}$ in the asymmetric unit. OmpK36$_{WT+TD}$ data to 1.5 Å were collected on I03 at Diamond Light Source at a wavelength of 0.97 Å using an Eiger2 XE 16M detector and processed using xia2 [53]. The resolution of both data was evaluated by half-dataset correlation coefficient in Aimless (cut-off less than 0.3) [54]. The space group was determined to be *P1* with six copies of OmpK36$_{WT+TD}$ in the asymmetric unit. Further processing was performed using the CCP4 suite [55].

Both the OmpK36$_{WT+TD}$ and OmpK36$_{WT+D}$ structures were determined by molecular replacement in Phaser [56] using the OmpK36$_{WT}$ structure (PDB ID: 6RD3) [16] as a search model. Refinement of both structures was carried out in Phenix [57]. After rigid body and restrained refinement electron density corresponding to the mutations and insertions were identified, built and refined. Additional density, possibly detergent or lipid molecules, that was observed on the surface of the protein was also modelled. The final OmpK36$_{WT+TD}$ model has an $R_{work}$ of 19% and an $R_{free}$ of 20.5%, and the OmpK36$_{WT+D}$ model has an $R_{work}$ of 18.5% and an $R_{free}$ of 21.3%, respectively.

The data collection and refinement statistics for both the OmpK36$_{WT+TD}$ and OmpK36$_{WT+D}$ crystals are summarised in **S2 Table**. The coordinates and structure factors of OmpK36$_{WT+TD}$ and OmpK36$_{WT+D}$ have been deposited to the Protein Data Bank with PDB ID codes 7PZF and 7Q3T, respectively.

## Liposome swelling assays

Liposome swelling assays were performed by reconstituting proteoliposomes with recombinant OmpK36 variants as previously described without any modifications [16].

## *In vitro* competition assays

Overnight cultures of the indicated KP strains were mixed 1:1 in an eppendorf (e.g. 200 μl KP36 $_{WT}$ and 200 μl KP36$_{WT+D}$); KP36 $_{WT}$ was fluorescently tagged (sfGFP introduced into the silent *glmS* genomic site) to allow identification during enumeration. 5 μl of the 1:1 mixture was inoculated into minimal M9 media (1x M9 salts, MgSO$_4$ 4 mM, CaCl$^2$ 0.1 mM) supplemented with glucose (0.4%), lactose (0.4%) or glucose and Cas-aminoacids (2The cultures were grown overnight at 37°C, 200 rpm. CFU enumeration was performed on LB agar plates grown overnight at 37°C; to distinguish the competing strains plates were UV transilluminated to determine colony fluorescence.

## Antimicrobial susceptibility testing

Minimum inhibitory concentrations (MICs) were determined in triplicate by reference broth microdilution according to the ISO standard (ISO 20776–1:2019, https://www.iso.org/standard/70464.html) using sterile 96-well plates (SARSTEDT, Germany). Antibiotic and inhibitors powders were from the following sources: avibactam (AOBIOUS, U.S.A.), aztreonam (United Biotech, India), cefiderocol (Shionogi, Japan), ceftazidime, imipenem, meropenem, relebactam and vaborbactam (Merck, Germany). Cation-adjusted Mueller-Hinton broth (MHB) (Thermo Fisher Scientific, U.S.A.) was used for all agents except cefiderocol, for which iron-depleted MHB (Shionogi) was used. Inhibitors were used at fixed concentrations of 4 mg/L (avibactam, relebactam) or 8 mg/L (vaborbactam). Results were read after incubation at 35±1°C in ambient air for 18±2 hours, and interpreted according to the EUCAST clinical breakpoints v 12.0, 2022 (https://www.eucast.org/clinical_breakpoints/), except for aztreonam/avibactam for which the EUCAST clinical breakpoint for aztreonam was used. *Escherichia coli* ATCC 25922, *Pseudomonas aeruginosa* ATCC 27853, *Klebsiella pneumoniae* ATCC 700603 and *K. pneumoniae* ATCC BAA-2814 were used as quality control strains according to EUCAST guidelines (https://www.eucast.org/ast_of_bacteria/quality_control/).

## Mouse infection

Female, BALB/c, 8–10 week old mice (Charles River, UK) were randomised into groups (co-housed in groups of 5) and acclimatised for one week before infection. Mice were housed under a 12 hour light dark cycle and had access to food and water *ad libitum*.

Inoculum was prepared from saturated overnight cultures grown in LB broth diluted in 1xPBS to a total volume of 50ul. This was delivered via intratracheal intubation (Kent Scientific) under ketamine (80mg/kg) and medetomidine (0.8mg/kg) anaesthesia. Monitored recovery from anaesthesia took place at 32–35°C following the administration of atipamezole (1mg/kg) reversal. Inoculum size was verified (+/-10%) by enumeration on agar plates.

Bacterial enumeration at experimental end-point took place on LB agar plates supplemented with 50mcg/ml rifampicin. Blood samples were collected at end-point by cardiac puncture under terminal anaesthesia (ketamine 100mg/kg, medetomidine 1mg/kg). Lungs were excised post mortem *en bloc* followed by homogenisation. Samples were 10-fold serially diluted before plating and agar plates incubated overnight at 37°C. In competition assays plates were UV transilluminated to determine colony fluorescence.

## Serum cytokine bead assay

Serum cytokine levels were determined using a beads-based immunoassay with a custom-designed mouse cytokine panel (LEGENDplex, BioLegend) following the manufacturer's instructions. Cytokine measurements were acquired using a FACSCalibur flow cytometer (BD

Biosciences), and flow cytometry analyses were performed using LEGENDplex data analysis software suite (https://www.biolegend.com/en-us/legendplex/software). All values were above the detection limit for both serum IL-6 and CXCL-1 (KC) levels and only 7/47 individual values (<15%; maximum of 2 per experimental group) were below the limit of detection when analysing TNF levels. These TNF values below the detection limit were assumed to be the lowest value detectable by the assay for statistical analysis and were displayed as 1 for easy visualisation in graphs (**Fig 5E**).

## Neutrophil quantification in lung homogenate by flow cytometry

1 ml of complete RPMI supplemented with penicillin and streptomycin solution (final concentration of 100 U and 100 μg/ml respectively), Liberase TM (0.13 mg/ml final, Roche) and DNaseI (10 μg/mL; Sigma-Aldrich) was added to the lung homogenate after removal of 30 μl for lung CFU calculations. The homogenate was then incubated for 40 mins on a shaker at 37˚C to allow for enzymatic tissue disruption and cell dissociation. After this incubation, the samples were placed on ice to interrupt the enzymatic digestion and the enzymes were further inhibited by adding EDTA (final concentration 10 mM; Gibco). The samples were run through a 70 μm cell strainer to obtain a uniform single-cell suspension, spun down and resuspended in 600 μl of complete RPMI. From here onwards, steps were carried out on ice to preserve cell viability.

For staining, ~5x10$^6$ cells (around 120 μl) from each sample were added per well in a 96-well V-bottom plate. Dead cells were routinely excluded with Zombie Aqua Fixable Dead Cell Stain (ThermoFisher Scientific). Single cell suspensions were incubated with Fc block (Miltentyi Biotec) in FACS buffer (1% BSA, 2mM EDTA in DPBS, Sigma), followed by staining with anti-CD11b-PerCP-Cy5.5 (#45-0112-82, ThermoFisher Scientific) and anti-Ly-6G-FITC (#551460, BD Pharmingen) in FACS buffer for 30 min, all at 4˚C in the dark. Fluorescence minus one (FMO) controls and a fully unstained sample were always included as controls. Stained cells were washed in FACs buffer and fixed for 20 mins at room temperature with 1% paraformaldehyde in PBS; fixed cells were kept in the dark at 4˚C until analysis, usually the following day. On the day of the analysis, single stain controls for compensation were prepared using VersaComp beads (Beckman Coulter) and ArC Amine Reactive Compensation beads (ThermoFisher Scientific) and a known volume of CountBright absolute beads (ThermoFisher Scientific) was added to each sample before running the samples. Flow cytometry analysis on 50000 live cells was performed on a BD LSRFortessa cell analyzer (BD Biosciences). Data were analysed using FlowJo software (Tree Star). Neutrophils were defined as CD11b + Ly6G+ live cells and absolute numbers of cells in the sample were calculated using the numbers of CountBright absolute beads counted following the manufacturer's instructions.

## Supporting information

**S1 Fig. L3 insertions have emerged across different *ompK36* gene backgrounds.** Midpoint-rooted phylogeny of intact *ompK36* genes from 14,888 KP isolates with highlighted nodes of all gene variants (**A**) and only those harbouring a specific L3 insertion type (**B-I**). (TIFF)

**S2 Fig. The D insertion codon and position have no impact on outer membrane (OM) abundance or meropenem MIC.** (**A**) The D insertion in OmpK36$_{WT+D}$ is mediated by an additional gac codon in the L3 coding region of OmpK36. We generated a codon switch mutant in which this additional gac was replaced with a synonymous gat codon and positional mutant in which the gac codon was moved -2 positions backwards, towards to 5' end of the

*ompK36* open reading frame (towards the N-terminal). (**B**) Outer membrane preparations (OMP) separated by sodium dodecyl sulfate–polyacrylamide gel electrophoresis (SDS-PAGE) and Coomassie staining demonstrating no change in OM OmpK36 abundance in with OmpK36$_{WT+D}$ in which the insertion is encoded by a gac or gat codon). (**C**) OMP followed by SDS-PAGE separation and Coomassie staining demonstrate no change in abundance when the D insertion is moved in position which functionally results in a non-significant change in the meropenem MIC (**D**). All strains in D have *ompK35* deleted and express KPC-2 from a pKpQIL-like plasmid.
(TIFF)

**S1 Table. See excel file.**
(XLSX)

**S2 Table. Data collection and refinement statistics.** Value in parenthesis refer to data in the highest resolution shell.
(TIFF)

**S3 Table. Primers used in this study.**
(TIFF)

## Acknowledgments

We would like to thank the Pathogen Informatics group from the Wellcome Sanger Institute for informatics support.

## Author Contributions

**Conceptualization:** Sophia David, Joshua L. C. Wong, Gad Frankel.

**Data curation:** Sophia David, Joshua L. C. Wong, Julia Sanchez-Garrido, Hok-Sau Kwong.

**Formal analysis:** Sophia David, Joshua L. C. Wong, Julia Sanchez-Garrido, Hok-Sau Kwong, Fabio Morecchiato, Tommaso Giani.

**Funding acquisition:** Konstantinos Beis, Gad Frankel.

**Investigation:** Sophia David, Joshua L. C. Wong, Julia Sanchez-Garrido, Hok-Sau Kwong, Wen Wen Low, Fabio Morecchiato, Tommaso Giani.

**Methodology:** Joshua L. C. Wong, Julia Sanchez-Garrido, Hok-Sau Kwong.

**Project administration:** Gad Frankel.

**Software:** Sophia David.

**Supervision:** Gian Maria Rossolini, Stephen J. Brett, Abigail Clements, Konstantinos Beis, David M. Aanensen, Gad Frankel.

**Validation:** Joshua L. C. Wong, Julia Sanchez-Garrido, Hok-Sau Kwong, Wen Wen Low, Fabio Morecchiato, Tommaso Giani.

**Visualization:** Sophia David, Joshua L. C. Wong, Julia Sanchez-Garrido, Hok-Sau Kwong.

**Writing – original draft:** Sophia David, Joshua L. C. Wong, Gad Frankel.

**Writing – review & editing:** Sophia David, Joshua L. C. Wong, Julia Sanchez-Garrido, Hok-Sau Kwong, Wen Wen Low, Fabio Morecchiato, Tommaso Giani, Gian Maria Rossolini, Abigail Clements, Konstantinos Beis, David M. Aanensen, Gad Frankel.

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
