## [Decision Letter · Decision Letter 0]

26 Mar 2022

Dear Prof. Frankel,

Thank you very much for submitting your manuscript "Widespread emergence of OmpK36 loop 3 insertions among multidrug-resistant clones of Klebsiella pneumoniae" for consideration at PLOS Pathogens. As with all papers reviewed by the journal, your manuscript was reviewed by members of the editorial board and by several independent reviewers. In light of the reviews (below this email), we would like to invite the resubmission of a significantly-revised version that takes into account the reviewers' comments.

An important concern raised in the reviews is about the novelty of the major findings where the results are to some extent confirmatory of the authors' previous work on the effect of GD insertions on porin structure as well as the fact that these insertions have been noted in other studies. Nevertheless, the novelty of the findings would be strengthened by a more extensive analysis of nutrients besides glucose that could have differential permeability between wild-type compared to L3 insertion mutants of OmpK36 strains hereby giving a mechanistic basis for difference in fitness. Ideally, a gut colonization experiment would address any concerns regarding transmission and would strengthen the impact of their findings. The authors need to address the concern regarding apparent conflicting statements regarding fitness of L3 insertion mutants.

We cannot make any decision about publication until we have seen the revised manuscript and your response to the reviewers' comments. Your revised manuscript is also likely to be sent to reviewers for further evaluation.

Sincerely,

Helena Ingrid Boshoff

Associate Editor

PLOS Pathogens

Raphael Valdivia

Section Editor

PLOS Pathogens

Kasturi Haldar

Editor-in-Chief

PLOS Pathogens

orcid.org/0000-0001-5065-158X

Michael Malim

Editor-in-Chief

PLOS Pathogens

orcid.org/0000-0002-7699-2064

An important concern raised in the reviews is about the novelty of the major findings where the results are to some extent confirmatory of the authors' previous work on the effect of GD insertions on porin structure as well as the fact that these insertions have been noted in other studies. Nevertheless, the novelty of the findings would be strengthened by a more extensive analysis of nutrients besides glucose that could have differential permeability between wild-type compared to L3 insertion mutants of OmpK36 strains hereby giving a mechanistic basis for difference in fitness. Ideally, a gut colonization experiment would address any concerns regarding transmission and would strengthen the impact of their findings. The authors need to address the concern regarding apparent conflicting statements regarding fitness of L3 insertion mutants.

Reviewer's Responses to Questions

**Part I - Summary**

Reviewer #1: The manuscript by David et al focuses on the Loop 3 of OmpK36 which has been shown to be implicated in resistance to carbapenems. They show that the main insertion in L3 tend to be TD, GD or D. They crystallize the protein and show that the pore size of these mutants is reduced, which allows them to be resistant to carbapenems, without affecting entry of glucose. They further show that these mutants do not have a defect as mono infections, but do tend to have a cost associated with them under co-infections. Lastly, they show that these mutants are still sensitive to new classes of drugs. Overall the manuscript is well written, and on an important topic of public health. The reviewer has one major and some minor concerns.

Reviewer #2: In this work, David et al study insertions in the outer membrane protein OmpK36 in K. pneumoniae. This study follows on previous work this group has published examining the structural and functional effects of L3 GD insertions in OmpK36. In the present work, the authors begin with a bioinformatics survey of OmpK36 sequences in public K. pneumoniae genomes. They find eight different classes of in-frame L3 insertions ranging in size from 1-3 amino acids present in approximately a quarter of a set of almost 15,000 porin sequences analyzed. Among these insertions, they established that GD was the most common in lineages in the Americas and Europe (concordant with previous smaller-scale assessments); D and TD insertions were the most common in Asia. The authors find that these (and other) insertions appear to be associated with large clonal expansions in the lineages of the hosting isolates, with most of the insertions in fact emerging multiple times (though the underlying nucleotide changes are surprisingly identical in most cases). The authors then present crystallographic studies of the D and TD insertions and characterize the resulting degrees of pore constriction. This is followed by meropenem MIC measurements on constructed insertion mutants in an appropriately prepared K. pneumoniae background strain. Next, they perform experiments to assess fitness cost associated with the insertions in a murine model. Finally, they measure MICs to a set of important new beta lactam/beta lactamase inhibitor combination antibiotics (CZA, AZT/AVI, IMP/REL, MER/VAB) for mutants containing the different insertions.

The study clearly represents a significant amount of work, and the experiments and analysis appear to be technically sound. Major issues discussed below.

Reviewer #3: David, Wong and colleagues present a comparative genomics and functional analysis of ompK36 L3 deletions in Klebsiella pneumoniae. ompK36 L3 deletions were found to have emerged multiple times, almost exclusively in K. pnuemoniae lineages harboring a carbapenemase. The authors go on to show that L3 deletions impact both pore structure (i.e. decreases size) and pore function (i.e. decreases diffusion of carbapenems). It is also noted that there is minimal impact of L3 deletions on the MICs to newer line combination therapy agents.

The authors also note that L3 deletions revert to WT alleles multiple times, suggesting that they can be associated with a fitness cost in certain unknown contexts. Based on competition assays in a mouse respiratory infection model, the authors hypothesize that decreased fitness in lungs may account for repeated reversion of L3 to wildtype.

Overall, this is an interesting study combining genomic and functional analyses to improve our understanding of the evolution of a set of mutations associated with increased antibiotic resistance. Moreover, both bioinformatic and experimental studies are rigorous and technically sound (with just a few technical issues raised below). My primary concern with this submission is repeated statements regarding the causal influence of ompK36 L3 deletions on clonal expansions of different lineages, when confounding mutations and observation bias in sequence database prevent such inferences being drawn from the analyses presented. In addition, there is over-confidence of results regarding the underlying selective pressures for gain and loss of L3 deletions, given the narrow testing of possible explanations.

**Part II – Major Issues: Key Experiments Required for Acceptance**

Reviewer #1: 1. Authors suggest a defect is only observed during in vivo competition. The single mutants colonize as WT, whereas in coinfection the WT outcompetes the mutant, potentially suggesting that there is a biological cost associated with L3 mutations that become apparent when there is competition for nutrients. To provide insight into this, authors should carry out competitive growth experiments in minimal media with either glucose or another carbon source. With a smaller pore size for the mutant it is possible that even though glucose can enter, its entry kinetics are different than WT, which would provide molecular reason for reversion to occur when antibiotic pressure is removed.

Reviewer #2: I found the discussion of fitness costs very confusing and apparently contradictory in places. Please see the quotes from different lines in the manuscript below:

Line 34: Author Summary “L3 insertions…impose only a low fitness cost”.

Lines 83-86: “We also demonstrate recurrent reversions of L3 insertions among clinical isolates, in line with the competitive disadvantage of L3 insertion mutants observed in our preclinical mouse pneumonia model in the absence of antibiotic therapy”.

Line 169-170 “L3 Insertions…revert at a low frequency”.

188-191 “Notably, the ST258/512 phylogeny also suggested that there have been multiple reversion events of L3 insertions, represented by genomes lacking a particular L3 insertion amidst a clade carrying that insertion (Figure 3B). The occurrence of reversions is suggestive of a selective pressure acting in favour of removing L3 insertions in certain contexts.”

Line 211 “The in vivo competitive disadvantage of L3 insertions explains the observed reversions”.

Line 231-233: “These experiments suggest that the L3 insertions do not significantly attenuate KP infection, thereby explaining the successful clonal expansions observed among isolates carrying these mutations.

Line 241-243: “These findings provide an experimental basis to explain the observed reversions of L3 insertions in the KP population as, whilst their expression supports a high capacity for infection, they result in a competitive disadvantage in the absence of antibiotics.”

I understand that there appears to be a genetic context-dependence of the fitness costs, and also that the fitness costs are more apparent in the murine competition experiment than in the single isolate infection experiments. But I think the findings need to be summarized in a way that appears consistent without blatant contradictions. If the conclusion is as in lines 188-191 that reversions are frequent and in lines 241-243 that there is a significant fitness disadvantage, then the Author Summary in Line 34 should not conclude “L3 insertions impose only a low fitness cost” and lines 169-170 should not state that L3 insertions…revert at a low frequency. I would suggest rewriting the sections pertaining to fitness costs so that they do not appear contradictory.

I also believe the authors need to do a better job with defining exactly what the novel reportable findings and biology are in this work. The bioinformatics study is expansive and well-done, but the main results are not really new as the most common insertions have been noted in many other studies. The structural analysis of the T and TD insertions is well done, but may represent a somewhat incremental advance over the authors’ prior work on the GD insertion and structural work on Omps done by other groups. There is also a more general literature on fitness costs of Omp mutations and deletions. I believe that adding some focusing sentences in each section that highlight the new findings (as opposed to those generally confirmatory of principles already understood) would be very helpful.

Reviewer #3: 1. Line 108 – It is stated based on the phylogenetic analysis in Figure S1 that the different L3 deletions emerged multiple times independently, suggesting that “these underlying mutations are more likely to evolve than alternatives”. First, there is no evidence presented that they are more likely to evolve, but perhaps more prone to spread. Second, while Figure S1 indicates that convergence is unlikely due to homologous recombination of the entire ompK36 gene, it does not rule out recombination of regions containing the L3 deletions. I would suggest exploring this possibility, as has been done recently for toxin allele switching in C. difficile (Mansfield et al., PLoS Pathogens, 2020).

2. On line 81 in the introduction, and repeatedly throughout the rest of the manuscript (e.g. line 200, line 204, etc.), it is stated that L3 deletions drove large clonal expansions of resistant clones of K. pneumoniae. However, there are several issues that prevent such a conclusion. First, as stated by the authors, the L3 deletions are linked to carbapenemase acquisition and also ompK35 deletions, making it impossible to isolate the specific effect of L3 deletions to lineage success. Arguing for the role of the carbapenemases themselves (or other genes/mutations co-acquired with them), there are several prominent lineages and sub-lineages that harbor those genes and have WT ompK36. A second issue with inferences regarding clonal expansions are severe observation biases within sequence databases for antibiotic resistant strains. In other words, it doesn’t seem possible to isolate the impact that ompK36 deletions had on lineage success, because studies often bias themselves by phenotypically screening for isolates that have these mutations. Finally, public databases have other types of sampling biases (e.g. over sequencing of outbreaks in certain geographies), which aren’t considered when making these claims.

3. To understand the overall stability of the ompK36 L3 deletions, along with their occasional convergent loss, the authors use a mouse pneumonia model. I have several concerns about this. First, the authors provide no justification for why this is a relevant model for providing insight into the transmission dynamics of Klebsiella mutants. Given that the presumed reservoir for transmission is the gut, it would seem a gut colonization model would be more appropriate. Second, the authors conclude that the lack of impact of L3 deletions on virulence explains why L3 deletions are stable. However, the rationale for this is unclear, as to my knowledge there is not a known association between severity of or propensity to cause respiratory infections, and epidemic success of K. pneumoniae lineages. Third, the authors state that the competitive advantage of WT versus L3 mutants in this respiratory model explains convergent reversion to WT in L3 mutant lineages. Again, there is no evidence that respiratory colonization/infection is the underlying selective pressure in human isolates. This would perhaps be more convincing if isolates with L3 reversions were enriched in respiratory isolation. Overall, these mouse experiments feel a bit like just so storytelling, where the outcome of the experiments are being interpreted to fit a story. However, there are a myriad of alternative hypotheses to explain selective pressures underlying proliferation and loss of L3 deletions were not explored.

4. When evaluating the function of porin mutants, why was glucose uptake measured? Are there other relevant nutrients taken up through this channel that might help account for fitness costs?

5. A bit more analysis on the genetic pathways for reversion of L3 deletions and their proliferation would be interesting. In particular, is there evidence that reversions are due to recombination mediated allele switching, and once these reversions occur, are they always dead ends?

**Part III – Minor Issues: Editorial and Data Presentation Modifications**

Reviewer #1: 1. Were any statistical differences observed in Fig 2F between the WT and mutants. The statistics are only shown between empty liposome and variants. The GD variant does have a small defect, which would again explain their in vivo competition experiment.

2. It is not clear from the manuscript, how the mutants were constructed. The reviewer assumes that the OmpK36 mutations are on the chromosome. The nomenclature used as an example OmpL36 WT+D is confusing as that implies if the WT allele is present and a second mutant allele is introduced. Authors can just refer to the modifications with subscript.

3. Also, the mutant alleles are initially mentioned in line 139, but only explained in 160-162. The authors can explain this when they first introduce the alleles.

4. Is OmpK35 deletion required for resistance? Do you require deletion/modification of both alleles? This is not addressed.

Reviewer #2: N/A

Reviewer #3: N/A

PLOS authors have the option to publish the peer review history of their article (what does this mean?). If published, this will include your full peer review and any attached files.

Reviewer #1: No

Reviewer #2: No

Reviewer #3: No
---

## [Editor Report · Decision Letter 1]

15 Jun 2022

Dear Prof. Frankel,

We are pleased to inform you that your manuscript 'Widespread emergence of OmpK36 loop 3 insertions among multidrug-resistant clones of Klebsiella pneumoniae' has been provisionally accepted for publication in PLOS Pathogens.

Best regards,

Helena Ingrid Boshoff

Associate Editor

PLOS Pathogens

Raphael Valdivia

Section Editor

PLOS Pathogens

Kasturi Haldar

Editor-in-Chief

PLOS Pathogens

orcid.org/0000-0001-5065-158X

Michael Malim

Editor-in-Chief

PLOS Pathogens

orcid.org/0000-0002-7699-2064

The authors have sufficiently addressed the reviewers' concerns.
---

## [Editor Report · Acceptance letter]

6 Jul 2022

Dear Prof. Frankel,

We are delighted to inform you that your manuscript, "Widespread emergence of OmpK36 loop 3 insertions among multidrug-resistant clones of Klebsiella pneumoniae," has been formally accepted for publication in PLOS Pathogens.

Best regards,

Kasturi Haldar

Editor-in-Chief

PLOS Pathogens

orcid.org/0000-0001-5065-158X

Michael Malim

Editor-in-Chief

PLOS Pathogens

orcid.org/0000-0002-7699-2064